# The simple-cubic structure of elemental Polonium and its relation to combined charge and orbital order in other elemental chalcogens

## Ana Silva and Jasper van Wezel⋆

Institute for Theoretical Physics, Institute of Physics, University of Amsterdam,
1090 GL Amsterdam, The Netherlands

⋆ vanwezel@uva.nl

## Abstract

Polonium is the only element to crystallise into a simple cubic structure under ambient conditions. Moreover, at high temperatures it undergoes a structural phase transition into a *less* symmetric trigonal configuration. It has long been suspected that the strong spin-orbit coupling in Polonium is involved in both peculiarities, but the precise mechanism by which it operates remains controversial. Here, we introduce a single microscopic model capable of capturing the atomic structure of all chalcogen crystals: Selenium, Tellurium, and Polonium. We show that the strong spin-orbit coupling in Polonium suppresses the trigonal charge and orbital ordered state known to be the ground state configuration of Selenium and Tellurium, and allows the simple cubic state to prevail instead. We also confirm a recent suggestion based on *ab initio* calculations that a small increase in the lattice constant may effectively decrease the role of spin-orbit coupling, leading to a re-emergence of the trigonal orbital ordered state at high temperatures. We conclude that Polonium is a unique element, in which spins, orbitals, electronic charges, and lattice deformations all cooperate and collectively cause the emergence of the only elemental crystal structure with the simplest possible, cubic, lattice.

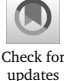

# 1 Introduction

Polonium is unique in the periodic table, being the only element to crystallise into a simple cubic lattice structure under ambient conditions. Besides it being remarkable that such a loosely packed configuration is favoured in any material, this is also surprising given that Tellurium (Te) and Selenium (Se), the two isoelectronic elements directly above Po in the periodic table, adopt a trigonal spiral lattice structure [1] (Sulphur and Oxygen in the same column form molecules rather than crystals, and will be ignored from here on). The trigonal arrangement in Te and Se can be understood as arising from a Peierls instability of a hypothetical simple cubic parent structure [2], in which the short bonds in three simultaneous charge density waves connect in a pattern that spirals around the body diagonal of the cube (see inset in figure 1).

Looking more closely, the spiral structure in Se and Te is in fact a combined charge and orbital ordered state, in which a spiral pattern of preferential occupation of different $p$-orbitals necessarily accompanies the charge order [2]. Polonium however, is considerably heavier than Se and Te, and relativistic effects may be expected to play a role in determining its ground state. Heuristically, it is clear that the presence of strong spin-orbit coupling, eliminating orbitals as individual degrees of freedom, is at odds with the formation of orbital order. In fact, *ab initio* calculations of the phonon dispersion in elemental chalcogens indicate that inclusion of relativistic effects suppresses a softening of the phonons, and possibly a related structural instability, which would otherwise be present [3–6]. The mechanism by which this is accomplished, as well as the identification of the dominant relativistic effect, being either a Darwin term, mass-velocity term, or atomic spin-orbit interaction, is still an unsettled and controversial issue [3–8]. In this paper, we construct a minimal microscopic model for elemental chalcogens, in which the evolution of the lattice structure can be studied as a function of the strength of spin-orbit coupling.

We show that at weak coupling, the simple cubic structure is unstable towards the formation of combined charge and orbital order, which results in the spiral trigonal lattice structure observed in Se and Te. Upon raising the strength of the spin-orbit coupling, the instability is suppressed, and the simple cubic structure observed in Po is realised instead. Moreover, we show that taking into account thermal expansion of the lattice, the strutural instability is suppressed at elevated temperatures. That is, using parameter values that are realistic for Po, the phonon structure is softened to such an extent as to effectively weaken the role of spin-orbit coupling at high temperatures. As a result, we find a transition between the two known allotropes of Polonium, the simple cubic $\alpha$−Po and the trigonal $\beta$−Po. We argue that this corresponds to the experimentally observed transition at approximatly 348$K$ [7,9], and conclude that like Se and Te, $\beta$−Po has a combined charge and orbital ordered structure (as indicated in the phase diagram of figure 2). The unusual lowering of the crystal symmetry upon raising temperature, and the peculiar phase diagram connecting the structure of Po to that of Se and Te, are thus found to be due to the intricate interplay between spins, orbitals, charges, and lattice deformations in the elemental chalcogens, where none of these degrees of freedom can be neglected.

# 2 Minimal microscopic model

The starting point for constructing a minimal microscopic model capable of describing the lattice instabilities in the entire family of elemental chalcogens, is a simple cubic arrangement of atoms. All chalcogens have four electrons in the outer shell of $p$-orbitals, so we will consider a tight-binding model taking into account only $p_x$, $p_y$, and $p_z$-orbitals on each site. For convenience, we choose the quantisation axes for the orbitals to coincide with the lattice directions.

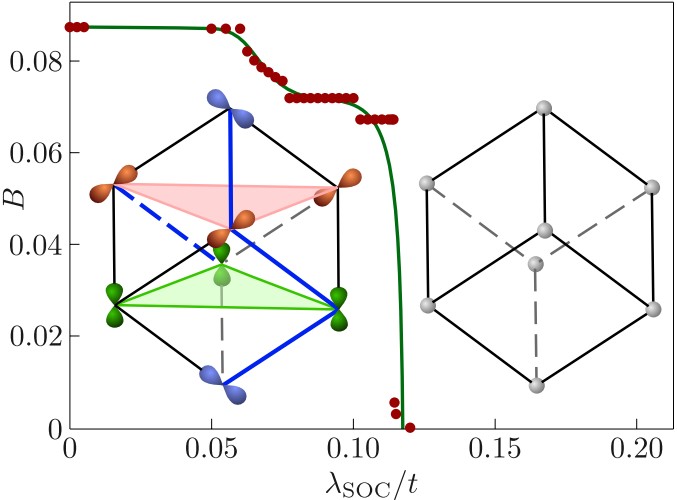

Figure 1: The value of the bond density order parameter $B$, as a function of the strength of spin orbit coupling $\lambda_{\text{SOC}}$, which is measured in units of the bandwidth $t$. The points are self-consistent numerical solutions of the set of equations in Eq. (4), while the solid line connecting them is a guide to the eye only. The unordered state at high $\lambda_{\text{SOC}}/t$ corresponds to the simple cubic lattice structure of $\alpha-$Po, as shown in the right inset. In the ordered state, the planes perpendicular to the cube's body diagonal move closer together, by means of a contraction of the thick bonds shown in blue in the left inset. The result is the spiral trigonal lattice structure known to be realised in Se and Te. Because the structural transition is the result of three simultaneous density wave instabilities, each occurring in chains of distinct orbitals, the trigonal state necessarily is also an orbital ordered state. The least occupied orbitals in each trigonal plane are highlighted in the left inset.

The strongest orbital overlaps then occur in one-dimensional chains of $p$-orbitals aligned in a head-to-toe fashion along their long axis. In other words, the overlaps of for example neighbouring $p_x$ orbitals on the $x$-axis are much larger than those between neighbouring $p_x$ orbitals on the $y$ or $z$ axes, or between any two $p$ orbitals of different type.

A minimal model for the bare electronic structure may thus be constructed by taking into account a hopping integral $t$ along chains in all three directions, but neglecting all other orbital overlaps, and in particular any inter-chain hopping. Interactions between one-dimensional chains in different directions can then be taken into account by including the Coulomb interaction $V$ between electrons in different $p$-orbitals on the same site. The resulting model is known to qualitatively capture the instability in the electronic structure which underlies the formation of combined charge and orbital order in Se and Te [2]. The electronic Hamiltonian for this minimal model can be written as the sum of tight binding and Coulomb terms:

$$\hat{H}_{\text{TB}} = t \sum_{\mathbf{r},n,\sigma} \hat{c}^\dagger_{\mathbf{r},n,\sigma} \hat{c}_{\mathbf{r}+\mathbf{n},n,\sigma} + \text{H.c.}$$

$$\hat{H}_{\text{Coul}} = V \sum_{\mathbf{r},n,\sigma,\sigma'} \hat{c}^\dagger_{\mathbf{r},n,\sigma} \hat{c}_{\mathbf{r},n,\sigma} \hat{c}^\dagger_{\mathbf{r},n+1,\sigma'} \hat{c}_{\mathbf{r},n+1,\sigma'} , \tag{1}$$

where $\hat{c}^\dagger_{\mathbf{r},n,\sigma}$ creates an electron on position $\mathbf{r}$, with spin $\sigma$, in a $p_n$-orbital, with $n \in \{x,y,z\}$. The lattice vectors $\mathbf{a}_n$ are written using the shorthand notation $\mathbf{n}$. In our simulations, we use the parameter values $t = 2.0\,\text{eV}$ and $V = 39\,\text{meV}$.

We additionally allow atoms to be displaced by introducing phonons. Since the phonon

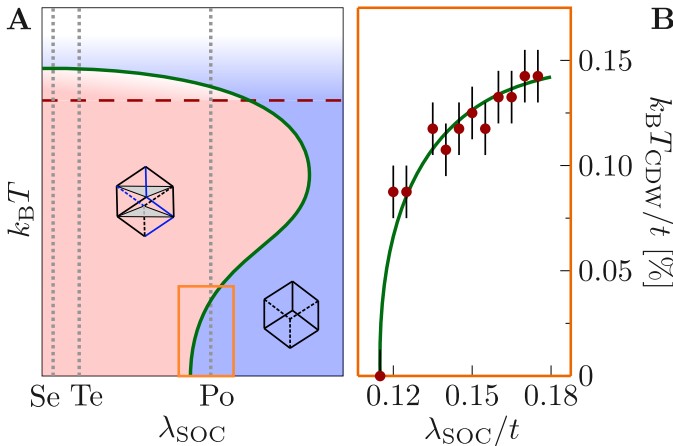

Figure 2: Phase diagram for elemental chalcogens as a function of temperature and strength of spin-orbit coupling. **A.** Schematic phase diagram. At low temperatures, increasing the strength of spin orbit coupling leads to a suppression of the trigonal instability, and hence a stabilisation of the simple cubic lattice. Polonium is expected to fall just to the right of the transition point, and thus to have a simple cubic ground state, while Selenium and Tellurium have low spin-orbit interaction, and thus a trigonal ground state structure. At fixed spin-orbit coupling, starting from the trigonal phase, the melting point (schematically indicated by the red dashed line) is encountered before charge and orbital order is destroyed and the local structure becomes cubic. Starting instead from the simple cubic phase, thermal expansion of the lattice lowers the bare phonon energy and thus shifts the balance of competing interactions in favour of the trigonal phase. **B.** The transition temperatures between cubic (lower right) and trigonal (upper left) phases, found by self-consistently solving the mean field equations. The error bars indicate the uncertainty in assigning the transition point within the precision of our numerical routine, and the solid line is a guide to the eye.

dispersion is approximately flat in the momentum-space region of interest, we employ an Einstein mode of constant energy $\hbar\omega = 3.5\,\text{meV}$. There are two different ways in which electrons couple to the phonons. On the one hand, atomic displacements alter the interatomic distances, which affects the hopping of electrons between them. On the other hand, atomic displacements also alter the local density of ions surrounding a particular site, which influences the on-site potential energy of electrons. We take into account both the kinetic and potential energy contributions of phonons:

$$\hat{H}_{\text{el-ph}}^{\text{kin}} = g^{(1)} \sum_{\mathbf{r},n,\sigma} \left( \hat{u}_{\mathbf{r},n} - \hat{u}_{\mathbf{r}+\mathbf{n},n} \right) \hat{c}_{\mathbf{r},n,\sigma}^{\dagger} \hat{c}_{\mathbf{r}+\mathbf{n},n,\sigma} + \text{H.c.}$$

$$\hat{H}_{\text{el-ph}}^{\text{pot}} = g^{(2)} \sum_{\mathbf{r},n} \left( \hat{u}_{\mathbf{r}+\mathbf{n},n} - \hat{u}_{\mathbf{r}-\mathbf{n},n} \right) \hat{c}_{\mathbf{r},n,\sigma}^{\dagger} \hat{c}_{\mathbf{r},n,\sigma}. \tag{2}$$

Here $\hat{u}_{\mathbf{r},n}$ is the operator corresponding to the $n$-component of displacement for the atom on position $\mathbf{r}$. The relative strength of the two types of electron-phonon coupling $g^{(1)}$ and $g^{(2)}$ determines whether a spiral trigonal structure consisting of site-centered or bond-centered charge density waves is formed in Se and Te. For simplicity, we assume equal values $g^{(1)} = g^{(2)} = 0.04\,\text{eV}$ for these couplings, resulting in a bond-centered spiral state consistent with experimental observations.

The minimal model consisting of the terms considered so far gives rise to three sets of

mutually parallel Fermi surface sheets. This situation is extremely well-nested, and, together with the electron-phonon coupling, renders the simple cubic phase unstable towards the formation of three simultaneous charge density waves, connected to the three sets of planes. In fact, a single, common nesting vector $\mathbf{Q} = 2\pi/3a(1,1,1)$ can be chosen such that every point on a Fermi surface sheet is connected to a corresponding point on a parallel sheet. The on-site Coulomb interaction provides a coupling between the density waves, resulting in an overall spiral trigonal structure. Because each charge density wave resides in chains of a particular type of orbital, the trigonal structure is automatically orbital ordered as well as charge ordered [2].

In Polonium, we expect relativistic effects to suppress the trigonal $\beta$-Po phase at low temperatures, and instead stabilise the simple cubic $\alpha$-Po allotrope. This is made possible in the minimal model by including spin-orbit coupling:

$$\hat{H}_{\mathrm{SOC}} = \lambda_{\mathrm{SOC}} \sum_{\mathbf{r},n,n',\sigma,\sigma'} M_{nn'\sigma\sigma'} \hat{c}^\dagger_{\mathbf{r},n,\sigma} \hat{c}_{\mathbf{r},n',\sigma'}, \tag{3}$$

where $\lambda_{\mathrm{SOC}}$ is the overall strength of the spin-orbit coupling, while $M$ contains the matrix elements of the operator $\hat{\mathbf{L}} \cdot \hat{\mathbf{S}}$ in the basis of states labelled by orbital index $n$ and spin $\sigma$.

The full Hamiltonian, combining all terms from equations (1), (2), and (3), and taking arbitrary but realistic values for all model parameters, can be solved numerically within the mean field approximation. This is done by introducing mean field averages corresponding to charge density, bond density, and displacement waves in each of the three lattice directions:

$$\sum_\sigma \langle \hat{c}^\dagger_{\mathbf{r},n,\sigma} \hat{c}_{\mathbf{r},n,\sigma} \rangle = \rho_0 + A \cos(\mathbf{Q} \cdot \mathbf{r} + \varphi_n)$$

$$\sum_\sigma \langle \hat{c}^\dagger_{\mathbf{r},n,\sigma} \hat{c}_{\mathbf{r}+\mathbf{n},n,\sigma} \rangle = \sigma_0 + B \cos(\mathbf{Q} \cdot (\mathbf{r}+\mathbf{n})/2 + \varphi_n)$$

$$\langle \hat{u}_{\mathbf{r},n} \rangle = \tilde{u} \sin(\mathbf{Q} \cdot \mathbf{r} + \varphi_n). \tag{4}$$

Here, $A$ is the mean-field amplitude for the on-site charge density variations, while $B$ corresponds to modulations of the bond densities. The atomic displacement field is given by $\tilde{u}$. The wave vector $\mathbf{Q}$ is equal for all instabilities and is determined by the strongly nested Fermi surface, but the phases $\varphi_n$ differ between density waves in different lattice directions $n$. Taking $\varphi_n = n \cdot 2\pi/3$, the known spiral trigonal lattice structure of Te and Se is recovered for vanishing spin-orbit coupling. This relation can be understood as an optimisation of the competition between Coulomb and electron-phonon interactions [2], and is assumed to hold also for finite values of the spin-orbit coupling.

The phonon part of the mean field Hamiltonian can be solved analytically using a Bogoliubov transformation [10], which shows the atomic displacements in the presence of given electronic order parameters $A$ and $B$ to be $\tilde{u} = 2\sqrt{3}/\hbar\omega(2Bg^{(1)} - Ag^{(2)})$. This expression relates the displacement field $\tilde{u}$ to the amplitudes of site-centered and bond-centered charge modulations. Notice that the size of displacements is inversely proportional to the bare phonon frequency. The fermionic part of the mean field Hamiltonian can be written in matrix form and diagonalised numerically for any given value of $\tilde{u}$. Iterating this procedure eventually yields self-consistent solutions for the displacement $\tilde{u}$ and the density modulations $A$ and $B$.

Without spin-orbit coupling and at zero temperature, the mean field ground state has a non-zero expectation value for the displacements, and is hence in the spiral trigonal lattice configuration. As the strength of spin-orbit coupling, $\lambda_{\mathrm{SOC}}$, is increased, a critical point is encountered, beyond which no non-trivial self-consistent solutions exist, as shown in figure 1. Intuitively, the disappearance of the trigonal state at large $\lambda_{\mathrm{SOC}}$ can be understood by realising that it corresponds to a state of combined charge and orbital order. The strong coupling

between spin and orbitals destroys the independent orbital degree of freedom, and hence prevents the onset of orbital order. As a result, the simple cubic lattice remains the ground state configuration. Alternatively, the competition between spin orbit coupling and density wave order may be phrased in terms of energetics. Large spin orbit coupling causes the Fermi surface to deform and gaps to open up. This obstructs the formation of charge and orbital order, which depends on having sufficiently nested Fermi surface available for a charge ordering gap to lower the overall electronic energy.

## 3 Turning up the temperature

It is known experimentally that polonium undergoes an unusual structural phase transition at about 348 K, where the low temperature simple cubic $\alpha-$Po lattice structure is reduced in symmetry and becomes the high temperature trigonal $\beta-$Po phase [9, 11, 12]. In order to describe this effect in our minimal model, we include the effect of temperature in two places. First, the mean field expectation values all become thermal expectation values, written for the electronic part of the Hamiltonian in terms of Fermi-Dirac distributions. Secondly, and more importantly, we take into account the fact that thermal expansion of the lattice will cause a lowering of the bare phonon energy. Owing to the relative softness of the material, the change in phonon energy in Po is significant, and cannot be neglected [6].

To describe the dependence of phonon energy on temperature, we first approximate the thermal expansion to be linear, so that the lattice constant at temperature $T$ can be written as $a(T) = a_0(1 + \alpha \Delta T)$. Here $a_0$ is the lattice constant at some reference temperature ($\Delta T = 0$), and $\alpha$ is the linear thermal expansion coefficient, which we take to be the experimentally determined value $\alpha = 23.5 \times 10^{-6} K^{-1}$, obtained at $298 K$ [13]. Taking $\alpha$ to be fixed while varying the temperature is seen to be a reasonable approximation in the region of interest by comparing it to the volumetric thermal expansion in Po as obtained by first principle calculations [5]. Assuming the phonon energy to depend on the interatomic distance, the expansion of the lattice will cause the bare phonon energies to soften, which we describe by the linear dependence:

$$\hbar\omega = \hbar\omega_0 + \gamma(a(T) - a_0)/a_0, \tag{5}$$

where $\hbar\omega_0$ is the energy of the bare phonon at the reference temperature where $\Delta T = 0$ and $a(T) = a_0$. Fitting equation (5) to experimental data in order to establish the value of $\gamma$ is prevented by the fact that polonium's strong radioactivity leads to a scarcity in relevant experimental data. A rough estimate of $\gamma \approx -172$ meV can nonetheless be obtained by fitting equation (5) to *ab initio* studies of phonon energy versus lattice constant, reported in reference [6].

The lattice expansion affects the fermionic part of the mean field calculations through the inverse proportionality of the displacement $\tilde{u}$ on the bare phonon energy. Looking for self consistent solutions as a function of both temperature and spin-orbit coupling then leads to the phase diagram shown in figure 2. At zero temperature, sufficiently large values of spin-orbit coupling are seen to effectively prevent the simple cubic structure from distorting into a trigonal phase. Raising the temperature lowers the bare phonon energy however, which makes the simple cubic structure more unstable, and hence requires ever larger spin-orbit coupling to prevent it from breaking down. As a result, for any fixed value of the spin-orbit coupling, the lattice may undergo a charge ordering transition into the trigonal charge and orbital ordered state, even if the low temperature phase was simple cubic. This effect is shown once more in figure 3 in terms of the thermal evolution of the order parameter for fixed values of the spin-orbit coupling. Notice that the predicted chirality of the combined charge and orbital ordered

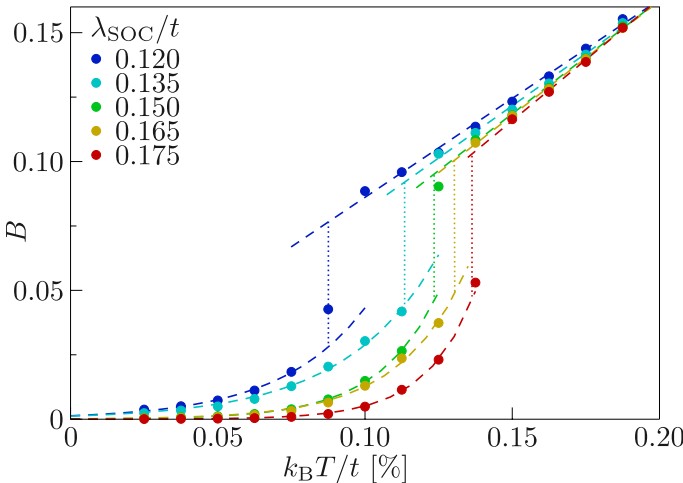

Figure 3: The value of the order parameter $B$ as a function of temperature, at various fixed values of the spin orbit coupling strength. At low temperature the lattice is simple cubic and order is exponentially suppressed (as indicated by the exponential fits to the data), while high temperatures favour the formation of combined charge and orbital order within a trigonal lattice structure (as shown by the linear fits). The transition into the ordered state, qualitatively indicated by the dotted lines, shifts to progressively higher temperatures for increasing strength of the spin-orbit coupling.

phase can in principle be observed in x-ray diffraction of optical activity experiments, while the orbital order itself should yield observable signatures in dedicated STM experiments.

The phase diagram of figure 2 agrees qualitatively with the evolution of lattice structures throughout the family of elemental chalcogens. The spin-orbit coupling in elemental Se and Te is weak enough to place them to the left of the zero-temperature transition point, as indicated schematically by the dashed lines in figure 2. Notice that at extremely high temperatures, the combined charge and orbital order in these crystals may be expected to be destroyed by thermal fluctuations. There is no guarantee however that this will happen below the melting temperature of the material. In fact, there are experimental indications that the short-range coordination in molten elemental Te changes from trigonal to cubic just above its melting temperature [14–16]. In contrast, polonium has strong spin-orbit coupling, placing it to the right of the zero-temperature transition, where the thermal evolution going from zero to high temperatures includes a transition from simple cubic to the less symmetric trigonal phase before the melting point is reached. Although probably impractical, further experimental exploration of the phase diagram of figure 2 could in principle be achieved by considering different isotopes of Po, in which the change in atomic mass affects the strength of the spin-orbit coupling.

## 4 Conclusions

The unique simple cubic lattice structure of elemental $\alpha-$Po at ambient conditions, as well as its unusual symmetry-lowering structural transition towards $\beta-$Po at elevated temperatures, can be qualitatively understood in terms of the minimal microscopic model presented here. That the lattice structures and phase diagrams of the isoelectronic elements Se and Te can be understood within the same model without any additional assumptions, firmly establishes the fact that it captures the essential physics in the description of crystalline elemental chalcogens.

The simple cubic ground state of polonium is found in this model to be of a deceptive simplicity. The electronic structure consists of well-nested pieces of Fermi surface, which in the presence of electron-phonon coupling inevitably lead to large peaks in the electronic susceptibility and hence an incipient structural instability. The fact that three separate instabilities loom in three distinct orbital sectors, coupled together by Coulomb interactions, yields a preferred trigonal configuration of the lattice, corresponding to a combined charge and orbital ordered state. This novel type of order is in fact realised in Se and Te, which have spiral trigonal lattice structures at all temperatures. In polonium however, the additional presence of strong spin-orbit coupling competes with the onset of charge and orbital order, which can be understood either in terms of the orbital degree of freedom becoming obsolete, or in terms of decreased nesting due to gaps opening up at the Fermi energy. The spin-orbit coupling thus prevents the simple cubic lattice from becoming unstable. At elevated temperatures, the balance is once again shifted in favour of the structural instability, by the softening of phonon energies as the lattice expands. The result is a re-emergence of the spiral trigonal state, but now at high temperatures, sitting above a more symmetric low-temperature simple cubic phase.

The elements Selenium, Tellurium, and Polonium, thus emerge as crystals in which an intricate balance between all possible degrees of freedom, orbitals, charge, spin, and atomic displacements, determines the structure of the atomic lattice. The fact that multiple degrees of freedom cooperate and compete with each other profoundly affects the physics of these deceptively simple materials, as can be clearly seen from the phase diagram across the family of chalcogens. Spin-orbit coupling competes with the onset of a cooperative charge and orbital ordered phase. This can be undone at high temperatures, but rather than thermal fluctuations determining the evolution of the phase diagram along the temperature axis, it is the indirect effect coming from the softening of phonons upon thermal expansion that shifts the balance of power between competing ingredients. That such a complex interplay can nonetheless be understood in terms of a simple minimal model, puts forward the family of chalcogens as a textbook case for understanding the possible effects of competition, co-existence, and cooperation among spin, charge, orbital, and lattice degrees of freedom.

**Funding information**   J.v.W. acknowledges support from VIDI grant 680-47-528, financed by the Netherlands Organisation for Scientific Research (NWO).

## A  Appendix: mean-field Hamiltonian

For completeness we present the mean-field Hamiltonian as obtained from equations (1)-(4). A momentum space basis may be defined as $(p_{x\uparrow}(k), p_{x\downarrow}(k), p_{y\uparrow}(k), \ldots p_{x\uparrow}(k+Q), p_{x\downarrow}(k+Q), \ldots p_{x\uparrow}(k-Q), \ldots)$. Here, the first index runs over the three types of $p$-orbitals, the second is a spin index, and the momentum is taken to lie within the reduced Brillouin zone. The Hamiltonian then has diagonal elements equal to $2t(\cos(k) - \mu)$ for the first six elements, $2t(\cos(k+Q) - \mu)$ for the next six, and $2t(\cos(k-Q) - \mu)$ for the final ones. The Coulomb interaction appears as elements of the form $VA(e^{\pm i\varphi_a} + e^{\pm i\varphi_b})$, connecting states with like orbitals and spins in different momentum sectors. The indices $a$ and $b$ correspond to the two orbital orientations different from the one of the states connected by this element. The electron-phonon coupling may be written as $-4e^{\pm i\varphi_a}[g^{(2)}A\sin(Q) - 2g^{(1)}B\sin(Q/2)][g^{(2)}\sin(Q) + g^{(1)}(\sin(k) - \sin(k'))]/(\hbar\omega)$. This term also connects states with like orbitals and spins, but different momenta. The index $a$ corresponds to the orbital index of the element under consideration, and the momenta $k$ and $k'$ are the momenta being connected. Finally, the spin orbit coupling acts within a momentum

sector, and is of the form:

$$\hat{H}_{\mathrm{SOC}} = \lambda_{\mathrm{SOC}} \begin{pmatrix} 0 & 0 & -i & 0 & 0 & 1 \\ 0 & 0 & 0 & i & -1 & 0 \\ i & 0 & 0 & 0 & 0 & -i \\ 0 & -i & 0 & 0 & -i & 0 \\ 0 & -1 & 0 & i & 0 & 0 \\ 1 & 0 & i & 0 & 0 & 0 \end{pmatrix}.$$

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
