# Peer review of "The simple-cubic structure of elemental Polonium and its relation to combined charge and orbital order in other elemental chalcogens"

_SciPost Physics, doi:SciPost Phys. 4, 028 (2018)_

## Round 2 · Referee Report · Anonymous (Referee 1) · 2018-2-16

Strengths

  1. Identifies an interesting problem -- an unusual transition in Polonium
  2. Provides a simple and plausible explanation for the lattice structure of chalcogen elements

Weaknesses

  1. Makes many assumptions -- e.g., electronic ordering drives lattice ordering, phonons interact in a simple way with electrons
  2. Does not make any testable predictions -- does not suggest how the validity of this picture can be tested

Report

The authors have presented a simple model to explain an unusual lattice transition that is seen in Polonium. The unusual aspect here is the lowering of symmetry above a critical temperature, from a simple cubic to a trigonal structure. The authors argue that the trigonal structure arises from electronic degrees of freedom via simultaneous charge and orbital ordering. In Polonium, they argue that spin-orbit coupling preempts this order. However, at higher temperatures, phonon softening (along with some other effects) brings back the instability to trigonal structure.

The paper represents a simple and plausible explanation for the lattice structures of chalcogens. This is interesting and worth publishing. However, there are many issues that are not clear.

  1. Some key aspects of the mean field scheme are not clearly described. What is the nesting wavevector Q? Are there multiple choices for Q? How do the authors select a single Q for all the orbitals?

  2. Was the mean field Hamiltonian constructed and diagonalised in momentum space or in real space? If it were a momentum space approach, it may not be possible to get a finite sized Hamiltonian matrix unless the vector Q is commensurate. If it were a real space approach, there may be strong finite size effects (the authors do not mention any finite size corrections).

  3. In Eq. 4, the authors impose a fixed phase relation between orbitals. In a consistent description of the physics, this is something that should emerge out of the calculation. To show that the approach is consistent, can the authors show that any other choice of the phases increases the energy?

  4. The dependence of phonon energy on temperature is, in principle, a highly non-linear quantity. The authors have used a linear form for this. Moreover, they say they have obtained the coefficient from fits to earlier ab initio data (Kang et al., PRB 2012).
    (a) The linear fit may only hold for a small temperature window. If a non-linear form is assumed, this may lead to quantitative corrections to the phase boundary. Perhaps, the qualitative picture will not change. (b) The authors have assumed an Einstein phonon dispersion, independent of temperature. The ab initio phonon band structures all show acoustic phonons. It is not clear how the authors managed to fit the data to extract a linear coefficient.

  5. This work presents a physical picture that is broadly consistent with experimentally known facts. It makes some key assumptions — the most important one being that lattice ordering is driven by electronic order. In order to make a concrete case, they should also propose some tests of their mechanism that can be checked by future experiments. I suggest that the authors suggest some qualitative tests. For example, they could discuss the effect of pumping in photons or a trend among isotopes of Polonium.

Requested changes

  1. The mean field Hamiltonian should be given explicitly, perhaps in an appendix.

The following are minor comments about the manuscript:

  1. The first line of the manuscript is an overstatement. The authors say “Polonium is unique in the periodic table, being the only element to crystallise into a simple cubic lattice structure.” The authors perhaps intend to say that Polonium is the only element to crystallise in a simple cubic structure in ambient conditions. There are several other elements which order in simple cubic geometry, e.g., alpha-Mn, beta-Mn, Cr, O and F.

  2. “ Moreover, we show that taking into account thermal expansion of the lattice, using parameter values that are realistic for Po, softens the phonon structure to such an extent as to effectively weaken the role of spin-orbit coupling in suppressing the structural instability.” — This is a run-on sentence that should be broken up into two or more sentences.

  3. “The full Hamiltonian,…, can be diagonalised numerically within the mean field approximation” — This can give the impression that the Hamiltonian is solved by exact diagonalisation. It may be better to say that “..can be solved numerically within the mean field approximation”.

---

## Round 3 · Author Response

We would like to sincerely thank the referee for their careful evaluation of our submission, and for their supportive comments and suggestions. We address the issues raised by the referee in the order in which they appear.

  • Strengths: we thank the referee for recognising the value of our work.

  • Weaknesses:

  • The referee points out that we make several assumptions.
 This is indeed true, and it leads to a very simple model for the family of elemental chalcogens. In fact, this should be seen as one of the main strengths of our approach. The experimentally observed structural phase diagram throughout a whole family of elements can be understood within a single, maximally simplified model. This is a clear indication that the model captures the essential physics. 

  • The referee claims that we make no testable predictions. 
In fact there are at least two clear and testable predictions already in the paper. One is the presence of orbital order in all elemental chalcogens. At the moment, this type of order is hard to detect experimentally. Dedicated STM setups or non-linear optical experiments however, could in principle detect the predicted orbital order. We point this out in the revised manuscript.
 The second prediction, is the phase diagram itself. As the referee already hints at in one of their later questions, looking at different isotopes of Po may be a way of tuning the value of spin-orbit coupling, which would allow a direct exploration of the predicted phase diagram.

- Report:
 We thank the referee for their concise summary and positive remarks.

  1. The referee asks about the nesting vector Q.
 The present manuscript builds on previous work done by the same authors, and published in Phys. Rev. B 97, 045151 (2018), where the possibility of a combined charge and orbitally ordered phase was first introduced. We therefore recognise that some aspects of the present work were only explained implicitly, by reference to our previous work. We have gone through the paper again, and added additional clarifications to address this issue.
 Concerning specifically the origin of the nesting vector Q: the model assumes as its starting point a simple cubic “parent” lattice, with a 2/3 filled band of p-orbitals. Considering only the dominant orbital overlap integrals, the simple cubic lattice consists of interwoven but independent one-dimensional chains running in all three lattice directions. The resulting electronic structure then contains three pairs of parallel planar Fermi surfaces. This situation is extremely well-nested, and a Peierls-type charge density wave is expected to emerge. In fact, a single nesting vector Q, corresponding to a body diagonal of the cube of intersecting Fermi surfaces, connects any point on the Fermi surface to a point on a parallel Fermi surface sheet. A single nesting vector can therefore gap the entire Fermi surface, and the dominant instability will be towards the formation of charge density waves in each of the three orbital sectors, sharing the same propagation direction Q = (2pi/3a, 2pi/3a, 2pi/3a). We summarise this discussion in the revised manuscript.

  2. The referee asks about the construction of the Hamiltonian.
 The tight-binding model was formulated in reciprocal space, and because the nesting vector is commensurate with the lattice, there is no problem with the size of the resulting Hamiltonian matrix. 
 We add a remark explaining this in the revised manuscript.

  3. The referee asks about the self-consistency of the fixed phase relations between the three CDW in our model.
 The mean-field equations in the present work were first solved self-consistently without spin-orbit coupling, keeping both the amplitudes and phases of the order parameter as free parameters. The reported phase relations for the self-consistent solutions are the lowest energy solutions found this way. We then assume the phase relations not to change significantly as a function of spin-orbit coupling.
 On the intuitive level, as long as the strength of spin-orbit coupling is sufficiently weak compared to the effects of the other interaction terms, the chiral trigonal lattice structure is expected to survive. Since this phase results from a competition between Coulomb and electron-phonon interactions, yielding combined charge and orbital order, it is consistent only with the fixed phase relations of eq (4). One may expect that at some critical value of the spin-orbit coupling the orbital order breaks down. At that point, rather than slightly modifying the phase relations, the parent simple cubic lattice is expected to emerge as the only self-consistent solution to the model. That no other stable phases exist was checked at several points in the phase diagram of figure 1.
 We extend the discussion of this point in the revised manuscript.

  4. a) The referee asks about the linear fit to the temperature dependence of the phonon energy.
 We agree with the referee that a linear fit can only hold true for a restricted window in temperature. It nevertheless suffices to gain a qualitative understanding of the observed structural phase diagram of Polonium, even if the results may not be quantitatively correct at temperatures far removed from the transition point.

b) The referee asks why we use an Einstein phonon mode in the model.
 The nesting vector Q in the chalcogens lies far from zero, so that even for acoustic modes the dispersion relation may be approximated to be constant (Einstein-like) in the momentum-space region of interest. 
 To account for the thermal evolution of the phonon mode, we assume the phonon energy to depend on the lattice constant, which in turn depends on temperature. Equation (5) then, is a first-order expansion of the phonon energy with varying lattice constant. Combining this with the linear thermal expansion of the lattice discussed above, yields the temperature dependence of the phonon energy. 
 We extend the discussion of this part of the analysis in the revised manuscript.

  1. The referee asks for qualitative tests of the presented model.
 The first clear experimental prediction coming out of this manuscript is that the trigonal lattice structure of beta-Polonium is in fact chiral, and orbital ordered. That is, it should be of precisely the same type as that observed in Selenium and Tellurium. For the latter two elements, the chirality of the lattice can be seen in X-ray diffraction as well as optical activity measurements. The orbital order is harder to detect experimentally at the moment. Dedicated STM setups or non-linear optical experiments, however, could in principle detect it. We point out these predictions in the revised manuscript. 
A second prediction could be based on the interesting suggestions of the referee. Looking at different isotopes of Po may be a way of tuning the value of spin-orbit coupling, which would allow a direct exploration of the predicted phase diagram. Making quantitative predictions in that direction, however, is beyond the scope of the current work.

  2. Requested changes: 
1. We now include the mean field Hamiltonian in an appendix, as suggested.

  3. We rephrased the first line of the manuscript in the suggested manner.
  4. We break up this sentence as suggested by the referee.
  5. We adopt the suggested formulation.

---

## Round 3 · List of Changes

See reply above

---

## Editorial Decision

published